# Paramagnetic encoding of molecules

Jan Kretschmer[1,2], Tomáš David [1], Martin Dračínský [1], Ondřej Socha [1], Daniel Jirak [3,4], Martin Vít [3,5], Radek Jurok[6,7], Martin Kuchař [6,8], Ivana Císařová [9] & Miloslav Polasek [1✉]

Contactless digital tags are increasingly penetrating into many areas of human activities. Digitalization of our environment requires an ever growing number of objects to be identified and tracked with machine-readable labels. Molecules offer immense potential to serve for this purpose, but our ability to write, read, and communicate molecular code with current technology remains limited. Here we show that magnetic patterns can be synthetically encoded into stable molecular scaffolds with paramagnetic lanthanide ions to write digital code into molecules and their mixtures. Owing to the directional character of magnetic susceptibility tensors, each sequence of lanthanides built into one molecule produces a unique magnetic outcome. Multiplexing of the encoded molecules provides a high number of codes that grows double-exponentially with the number of available paramagnetic ions. The codes are readable by nuclear magnetic resonance in the radiofrequency (RF) spectrum, analogously to the macroscopic technology of RF identification. A prototype molecular system capable of 16-bit (65,535 codes) encoding is presented. Future optimized systems can conceivably provide 64-bit (~10^19 codes) or higher encoding to cover the labelling needs in drug discovery, anti-counterfeiting and other areas.

[1] Institute of Organic Chemistry and Biochemistry of the CAS, Flemingovo náměstí 542/2, 160 00 Prague 6, Czech Republic. [2] Department of Organic Chemistry, Faculty of Science, Charles University in Prague, Hlavova 2030/8, 128 43 Prague 2, Czech Republic. [3] MR Unit, Department of Radiodiagnostic and Interventional Radiology, Institute for Clinical and Experimental Medicine, Vídeňská 1958/9, 140 21 Prague 4, Czech Republic. [4] Institute of Biophysics and Informatics, First Faculty of Medicine, Charles University in Prague, Salmovská 1, 120 00 Prague 2, Czech Republic. [5] Faculty of Mechatronics Informatics and Interdisciplinary studies, Technical University of Liberec, Hálkova 917/6, 460 01 Liberec, Czech Republic. [6] Forensic Laboratory of Biologically Active Substances, Department of Chemistry of Natural Compounds, University of Chemistry and Technology Prague, Technická 5, 166 28 Prague 6, Czech Republic. [7] Department of Organic Chemistry, University of Chemistry and Technology Prague, Technická 5, 166 28 Prague 6, Czech Republic. [8] Department of Experimental Neurobiology, National Institute of Mental Health, Topolová 748, 250 67 Klecany, Czech Republic. [9] Department of Inorganic Chemistry, Faculty of Science, Charles University in Prague, Hlavova 2030/8, 128 43 Prague 2, Czech Republic. ✉email: miloslav.polasek@uochb.cas.cz

Managing digital information is crucial not only for our civilization but for the existence of life itself. All known life forms depend on nucleic acids, which are in essence digital medium. The information capacity of molecules is enormous and very tempting for technological applications. Indeed, humans learned to hack the system of nucleic acids for the purpose of encoding data[1–7]. However, the principle of molecular recognition that works well within a biological microenvironment is difficult to connect with technology built primarily around the electromagnetism of inorganic materials and RF communication. Thus, alternative synthetic polymers have been proposed for information encoding[8–12]. Unfortunately, the current methods of reading often result in the destruction of the molecules. In contrast, modern civilization demands to communicate information wirelessly and repeatably, in an effort to identify and track objects[13,14], monitor health[15,16], and eventually merge the digital with the living[17,18]. To achieve this at the molecular level will require adapting the molecular systems to be more compatible with macroscopic information technologies.

Writing (encoding) information at the molecular level can be achieved with either (i) sequences of monomers concatenated within one molecular string[7–12] or with (ii) mixtures of individual, unique compounds[19–23]. For both, there is a trade-off between synthetic difficulty, material demands, and information capacity. Long sequences can be made from a few monomers, but synthetic writing is difficult, slow, and prone to errors[7–9]. On the other hand, mixtures are easy to make but require a high number of uniquely distinguishable components, which eventually become a limiting factor[19–23]. Theoretically, an alternative approach, better balancing the need to generate many codes with a small number of components and synthetic steps, is (iii) to first synthesize short sequences and then use them as unique components in mixtures. Although this approach is conceptually well-understood[21], the difficulty of decoding mixed sequences is prohibitive.

Reading (decoding) nongenomic molecular codes currently heavily relies on mass spectrometry (MS), which can decode sequences as well as complex mixtures with high sensitivity[8–10,19–21]. However, mixtures of sequences containing permutations of the same monomers have the same molecular weight and are nearly impossible to decode. In addition, MS is a destructive method that does not allow for repeated readings that are desirable for long-term tracking and monitoring. On the other hand, spectroscopic methods which are capable of this can typically only detect individual components (monomers) but cannot distinguish their order in sequences[12,24]. Another challenge for spectroscopic methods are signal overlaps that limit the number of distinguishable codes. For example, fluorescent labels, as one of the most important tools for interrogation of biological systems, provide relatively broad signals within a narrow spectral window[23–26]. Up to ten different fluorescent tags can be distinguished from mixtures with advanced computational methods, but this still amounts only to $2^{10} - 1 = 1023$ codes[26]. This limitation cannot be overcome without expanding the size along at least one dimension[23,27]. Indeed, various microscale particles have been proposed as barcodes, based on tuneable luminescent materials combined to generate the code[28–31]. However, the information capacity per mass is much lower for particles compared to molecules. A potential way out of this dilemma is to use non-optical methods of reading. For example, the spectral window of nuclear magnetic resonance (NMR) spectroscopy can accommodate much higher number of distinguishable signals usable for information encoding[22].

In the same way that magnetic media accelerated modern information technologies, it is recognized that molecular magnetism could be a major breakthrough for future technologies at the molecular scale. Alas, single-molecule magnets so far function only at impractically low temperatures[32]. The next best magnetic alternative that works at the molecular level and over a broad temperature range simultaneously is paramagnetism. When placed within an external magnetic field, unpaired electrons of a paramagnetic metal ion generate their own local field described by a magnetic susceptibility tensor. This tensor has specific radial and angular dependencies dictated by the particular element and its coordination environment, and strongly influences the NMR frequencies of nearby nuclei[33–35], an effect exploited in protein structure elucidation[36]. Lanthanides are ideally suited for this purpose, as they provide a range of chemically similar but magnetically different $Ln^{3+}$ ions amenable to stable incorporation into organic molecules via coordination[35,37]. While distinguishable NMR signals can be obtained from chemically different diamagnetic molecules and formulations[38–42], introducing paramagnetic lanthanides allows for decoupling of the NMR shifts from chemical properties. This principle has been previously used to create chemically nearly identical molecules that could be spectrally resolved with magnetic resonance imaging (MRI)[43–45]. However, with one lanthanide ion per molecule, the number of distinguishable signals remains limited to 12 usable lanthanides (excluding Pm, Gd; including 1 diamagnetic). So far, only a few works have investigated the possibility of combining paramagnetic effects of multiple lanthanides within one molecule[46,47], presumably due to the difficulty in synthesizing well-defined multimetallic compounds[24,48,49]. We speculated that two or more lanthanide ions arranged in sequence within one molecule should combine their magnetic susceptibility tensors to produce a unique NMR-readable outcome for each permutation of the elements. This greatly increases the number of signals and molecular codes that can be generated with a limited number of elements (Fig. 1).

In this work, we present a prototype of paramagnetically encodable molecular architecture, demonstrate the encoding and decoding principles, provide practical examples of information encoding, and outline the possible future capabilities. The number of unique codes that can be generated in this way from a few chemical building blocks is compatible with the foreseeable needs in drug discovery[1,2] and anti-counterfeiting[13] applications. This work provides an alternative direction toward programmable digital molecular information systems.

## Results

**Design of building blocks**. Combining magnetic susceptibility tensors of lanthanide ions requires their placement in close proximity within one molecule, preferably in a modular way to enable encoding of specific sequences. Conformational freedom and isomerism must be limited, so that the metal ion positions are well defined and there is no averaging or multiplication of NMR signals. Peptides can offer the desired modular synthetic approach, provided that a suitable building block is available to act simultaneously as amino acid and a strong chelator for $Ln^{3+}$ ions. Several amino acid/chelator building blocks have been described previously, but none were suitable for the task, falling short of the required control over conformations and/or isomerism (Fig. 2)[24,49–51]. Learning from these examples, we developed a family of building blocks collectively named DO3A-Hyp, where the chelator is based on macrocyclic DO3A (1,4,7,10-tetraazacyclododecane-1,4,7-triacetic acid) and the amino acid is derived from hydroxyproline (Hyp). Here, we present two members of the family, building blocks $L^1$ and $L^2$ (Fig. 2 and synthetic scheme in Fig. 3), that differ in the configuration of chiral centers in the Hyp moiety. The DO3A-Hyp family is structurally related to the chelator HP-DO3A that is clinically used in the MRI contrast agent *Gadoteridol*[37]. Although knowing

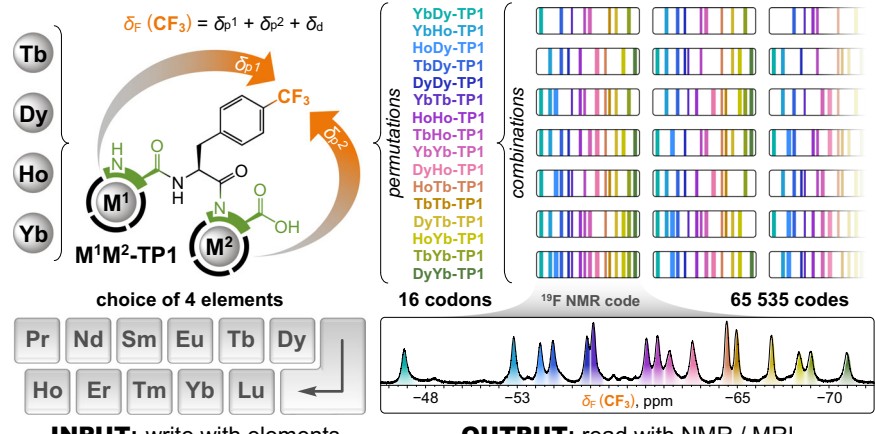

**Fig. 1 The principle of paramagnetic encoding presented in this work.** A sequence of two lanthanide ions ($M^1M^2$) is chemically written into a tripeptide molecule **$M^1M^2$-TP1**, which incorporates two stereochemically well-defined amino acid/chelator building blocks. The paramagnetic effects of the lanthanides combine to influence a reporter $CF_3$ group. Each $M^1M^2$ permutation of the elements (codon) results in a unique $^{19}F$ NMR shift ($\delta_F$) that is given by the sum of individual paramagnetic shift contributions from each $M^1$ and $M^2$ ($\delta_p^1$ and $\delta_p^2$, respectively) and a constant diamagnetic shift ($\delta_d$). The encoded molecules are further multiplexed to produce codes readable by NMR or MRI. A small number of elements allows the writing of a very high number of codes. The exact chemical structures of the amino acid/chelator building blocks used in this work are provided in Figs. 2 and 3.

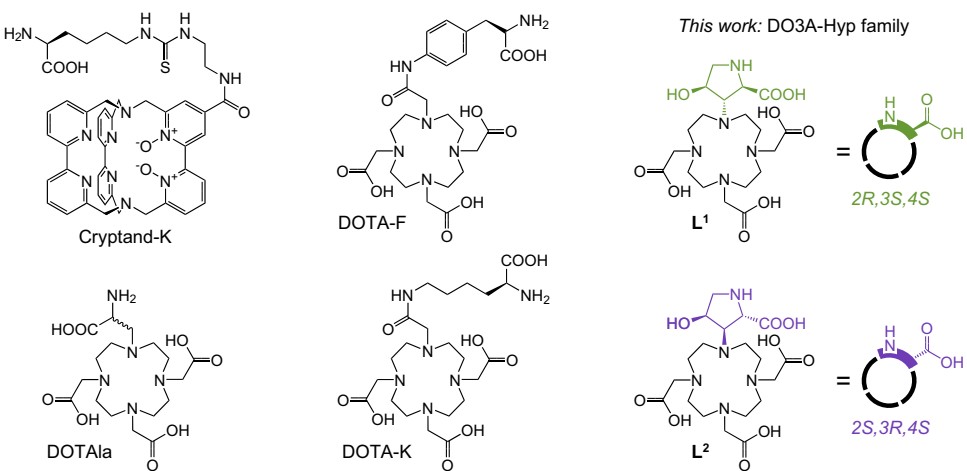

**Fig. 2 Chemical structures of amino acid/chelator building blocks for incorporation of $Ln^{3+}$ ions into peptides discussed in this work.** Structures were redrawn without protective groups and metal ions from references: Cryptand-K[24], DOTA-K[49,50], DOTA-F[50], and DOTAla[51]. The DO3A-Hyp family of building blocks presented in this work comprises two isomers that differ in the configuration of chiral centers in the Hyp (hydroxyproline) moiety: isomer *2R,3S,4S* (**$L^1$**, green shading) and isomer *2S,3R,4S* (**$L^2$**, violet shading).

the exact structure of the chelates was not necessary for the purpose of paramagnetic encoding, we found with X-ray diffraction on a $[Dy(L^1)]$ crystal that inversion ($2S{\to}2R,3S,4S$) occurred in Hyp during synthesis of $L^1$ (Fig. 4A and Supplementary Figs. 1–3). We explain this by deprotonation of the epoxide **5** during its reaction with basic cyclen (suggested mechanism in Supplementary Fig. 4). The same reaction step also opened a path to the isomer $L^2$, though with a much lower yield (9:1 ratio $L^1$: $L^2$). The crystal structure of $[Dy(L^2)]$ confirmed $2S,3R,4S$ configuration of Hyp in $L^2$ (Fig. 4B and Supplementary Figs. 5–7). In both solid-state structures, the Hyp moiety coordinated with the *2R-* or *2S-*carboxyl, respectively, rather than with the *4S-*hydroxyl group. Nevertheless, the *2R-*carboxyl of $[Ln(L^1)]$ chelates was reactive in peptide coupling reactions (in contrast to unreactive coordinated acetate arms), indicating coordination competition with *4S-*hydroxyl in solution (Supplementary Fig. 8). Coordination of *4S-*hydroxyl is therefore considered in figures to respect this reactivity. Our assumption was that the chiral Hyp moiety would impose a preference for one of several possible

coordination isomers typical for macrocyclic chelators and thus manage the problem of isomeric speciation[36,52,53]. Indeed, only one dominant species was found with NMR spectroscopy in solution for all tested lanthanides (see the next section).

**Encodable molecular framework.** Having the suitable building blocks, we designed a prototype of an encodable molecular framework. Tripeptide 1 (**TP1**) consists of two $L^1$ units, one at each terminus, and a middle amino acid bearing a reporter $CF_3$ group. Analogous tripeptide 2 (**TP2**) molecule contains $L^2$ unit at the C-terminus. This construction allows the reporter to perceive magnetic fields from two metal ions simultaneously in **$M^1M^2$-TP1** or **$M^1M^2$-TP2** molecules (Fig. 5). We selected the $CF_3$ reporter for the high sensitivity and negligible natural background of the $^{19}F$ nucleus, which facilitate observation and interpretation of signals in the RF NMR channel[40,41,53,54].

The chemical shifts induced by paramagnetic lanthanide ions are difficult to predict from theory with confidence[35], especially if the structure in solution is not precisely known. Therefore, to

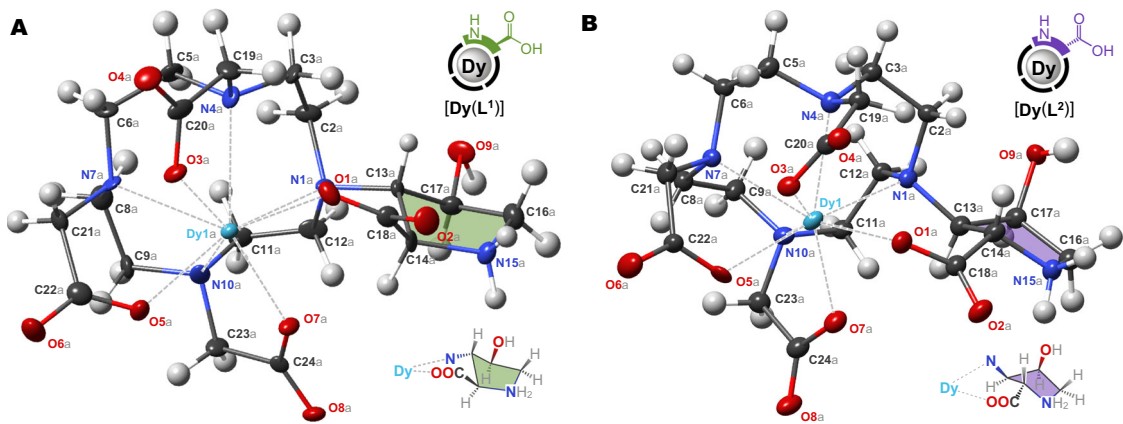

**Fig. 3 Synthetic scheme for building blocks L$^1$, L$^2$, and their protected variants.** Conditions: (i) BnBr (benzyl bromide), TEA (triethylamine), THF (tetrahydrofurane); (ii) PPh$_3$, THF, 0 °C followed by DIAD (diisopropyl azodicarboxylate), MeI, 0 °C → RT; (iii) (PhSe)$_2$, EtOH, NaBH$_4$, 0 °C; (iv) H$_2$O$_2$, THF, 0 °C → RT; (v) MCPBA (*meta*-chloroperoxybenzoic acid), CHCl$_3$, 85 °C; (vi) cyclen, *t*-BuOH, 105 °C; (vii) *t*-BuO$_2$CCH$_2$Br, K$_2$CO$_3$, MeCN; (viii) H$_2$, Pd@C, AcOH, MeOH; (ix) TFA; (x) FmocCl, aq. borate/NaOH buffer (pH 9.0), MeCN; (xi) MeO$_2$CCH$_2$Br, K$_2$CO$_3$, MeCN; (xii) Ac$_2$O, TEA, DMAP (*4*-dimethylaminopyridine), MeCN. Intermediates in brackets were not isolated. Colors define stereochemistry of the hydroxyproline moiety: *2R,3S,4S* (green) and *2S,3R,4S* (violet). Detailed synthetic procedures are provided in Supplementary Figs. 22–36.

**Fig. 4 Molecular structures of [Dy(L$^1$)] and [Dy(L$^2$)] in the solid state.** In both structures, the ligand coordinated to the central Dy$^{III}$ cation with four ring nitrogen donors and four carboxylate donors. The view highlights the absolute configuration of the proline ring. Thermal ellipsoids were set at 50% probability. **A** [**Dy(L$^1$)**] contains *2R,3S,4S* configuration of the proline ring (green shading) and the chelate adopts Δλλλλ square antiprismatic (SA) conformation. For detailed information, see Supplementary Figs. 1–3. **B** [**Dy(L$^2$)**] contains *2S,3R,4S* configuration of the proline ring (violet shading) and the chelate adopts Δδδδδ twisted square antiprismatic (TSA) conformation. For detailed information, see Supplementary Figs. 5–7.

gain a quick overview, we used a metal-free **TP1** for post-synthetic complexation of equimolar binary lanthanide mixtures to obtain statistical mixtures of four **M$^1$M$^2$-TP1** compounds in 1:1:1:1 molar ratio (Fig. 5B). In all studied cases, $^{19}$F NMR spectra showed four major singlets, mostly free of overlaps, shifted and broadened by the paramagnetic action of the respective Ln$^{3+}$ ions (Supplementary Fig. 9). We selected the Dy$^{3+}$/Ho$^{3+}$ combination for further detailed study because of a good balance between chemical shift and NMR signal broadening (Fig. 6A). Specific

M$^1$M$^2$ sequences were encoded into **M$^1$M$^2$-TP1** compounds via controlled synthesis using several different strategies, some of which also included direct peptide coupling with metal-preloaded derivatives of **L$^1$** (Supplementary Figs. 64–86). Each of the well-defined **M$^1$M$^2$-TP1** compounds matched one of the four major $^{19}$F NMR peaks observed in the statistical Dy$^{3+}$/Ho$^{3+}$ mixture, confirming that each M$^1$M$^2$ sequence generates a unique signal (Fig. 6B). Some minor peaks detected in the $^{19}$F NMR spectra of (purified) **M$^1$M$^2$-TP1** compounds could be attributed to low-

**Fig. 5 Synthesis of M¹M²-TP1 and M¹M²-TP2 compounds.** **A** Main synthetic pathway to well-defined **M¹M²-TP1** products. **B** Post-synthetic complexation pathway to statistical mixtures of four major **M¹M²-TP1** products. **C** Synthetic pathway to well-defined **M¹M²-TP2** products (analogous to A). Conditions: (i) Fmoc-Phe{*p*-CF₃}-OH, PyAOP ((7-azabenzotriazol-1-yloxy)tripyrrolidinophosphonium hexafluorophosphate), DIPEA (*N,N*-diisopropylethylamine), DMSO (dimethyl sulfoxide) followed by DBU (1,8-diazabicyclo[5.4.0]undec-7-ene), DMF (dimethylformamide); (ii) **Ac-Fmoc-Me₃L¹**, PyAOP, DIPEA, DMSO followed by TFA (trifluoroacetic acid); (iii) M²Cl₃, aq. MOPS (3-(*N*-morpholino)propanesulfonic acid)/NaOH (pH 7.0) followed by LiOH, H₂O, MeOH; (iv) M¹Cl₃, aq. MOPS/NaOH (pH 7.0); (v) LiOH, H₂O, MeOH; (vi) M¹Cl₃, M²Cl₃, aq. MOPS/NaOH (pH 7.0). For the full structures of the building blocks, see Fig. 3. Detailed synthetic procedures including alternative synthetic pathways are provided in Supplementary Figs. 37–95.

abundance forms of the same compound (conformers or isomers), because most of them coalesced with the main peak at elevated temperatures (Supplementary Fig. 10). Nevertheless, these minor peaks presented no obstacle in further analysis of the NMR data. As expected, the induced chemical shifts strongly depended also on the precise coordination environment and mutual arrangement of the two metal centers. The molecular system of **M¹M²-TP2**, despite its (deceptive) similarity to **M¹M²-TP1**, provided dramatically different chemical shifts with the same metal ion sequences (Fig. 6C and Supplementary Fig. 11). Furthermore, to investigate the importance of rigid and compact molecular design on paramagnetic encoding, we performed a control experiment using an alternative architecture. The same Dy³⁺/Ho³⁺ sequences were encoded into a tripeptide **TP3** that was constructed with the previously described DOTA-K building block[49,50] containing a flexible sidechain linker (Supplementary Fig. 12). Conformational flexibility, in this case, resulted in metal ion position averaging, diminishing the paramagnetic shifts, and consequently the **TP3** system failed to produce distinguishable signals for different M¹M² sequences (Fig. 6D and Supplementary Fig. 13). Moreover, the signals appeared split, likely due to the cis–trans isomerism of the coordinated amide moiety noted previously for similar compounds[36]. This problem was not observed with the **TP1** and **TP2** systems using the DO3A-Hyp family building blocks, demonstrating that a rigid and stereochemically well-defined molecular framework is essential for the purpose of paramagnetic encoding.

**Rules of paramagnetic encoding.** Various molecular systems have been previously explored that used a single lanthanide ion per molecule to generate discernible NMR shifts, primarily for imaging applications[43–45]. However, the way that magnetic susceptibility tensors of multiple lanthanide ions combine within one molecule is a rarely studied phenomenon[46,47]. We, therefore,

examined the rules of this in the **M¹M²-TP1** system in detail. Our hypothesis was that the observed ¹⁹F chemical shift ($\delta_F$) was a sum of three independent parameters:

$$\delta_F = \delta_p^1 + \delta_p^2 + \delta_d, \qquad (1)$$

where $\delta_p^1$ and $\delta_p^2$ are individual pseudocontact paramagnetic shifts generated by metal M in a given position (1 or 2, as defined in Fig. 1), and $\delta_d$ is a constant diamagnetic contribution.

To test this hypothesis, we first analyzed a limited dataset comprised of Dy³⁺/Ho³⁺, Dy³⁺/Y³⁺ and Ho³⁺/Y³⁺ statistical mixtures of **M¹M²-TP1** compounds. The diamagnetic Y³⁺ ion was included to anchor the $\delta_d$ component. The assignment of peaks to specific M¹M² sequences was already partially known (Fig. 6B) and the rest was inferred or determined by trial and error. A multiparametric least-square fit according to Eq. (1) converged to a single set of $\delta_p^1$, $\delta_p^2$, $\delta_d$ values. An independent dataset of Dy³⁺/Tm³⁺, Dy³⁺/Lu³⁺, Tm³⁺/Lu³⁺ mixtures provided similar $\delta_p^1$, $\delta_p^2$ values for Dy³⁺ that was present in both datasets and nearly identical $\delta_d$ (Supplementary Fig. 14). These results demonstrated that the contributions of individual paramagnetic centers were independent and additive, confirming the hypothesis. Knowing the parameters for Dy³⁺, Ho³⁺, and Tm³⁺ allowed us to decipher $\delta_p^1$, $\delta_p^2$ values for the remaining tested lanthanides. Final global fit provided very good agreement between the calculated and measured shifts, with residual differences not exceeding 0.34 ppm within a 38.8 ppm overall range (Supplementary Fig. 15). Because paramagnetic shifts are very sensitive to changes in molecular structure, this agreement demonstrates that the **M¹M²-TP1** compounds were practically isostructural regardless of the inserted lanthanide ions. For each element, the $\delta_p^1$, $\delta_p^2$ values were of opposite signs (Supplementary Fig. 15B), indicating substantial mutual rotation of the two magnetic susceptibility tensors in **M¹M²-TP1** molecule[35]. The $\delta_p^1$ and $\delta_p^2$ values roughly followed tabulated Bleaney constants[33],

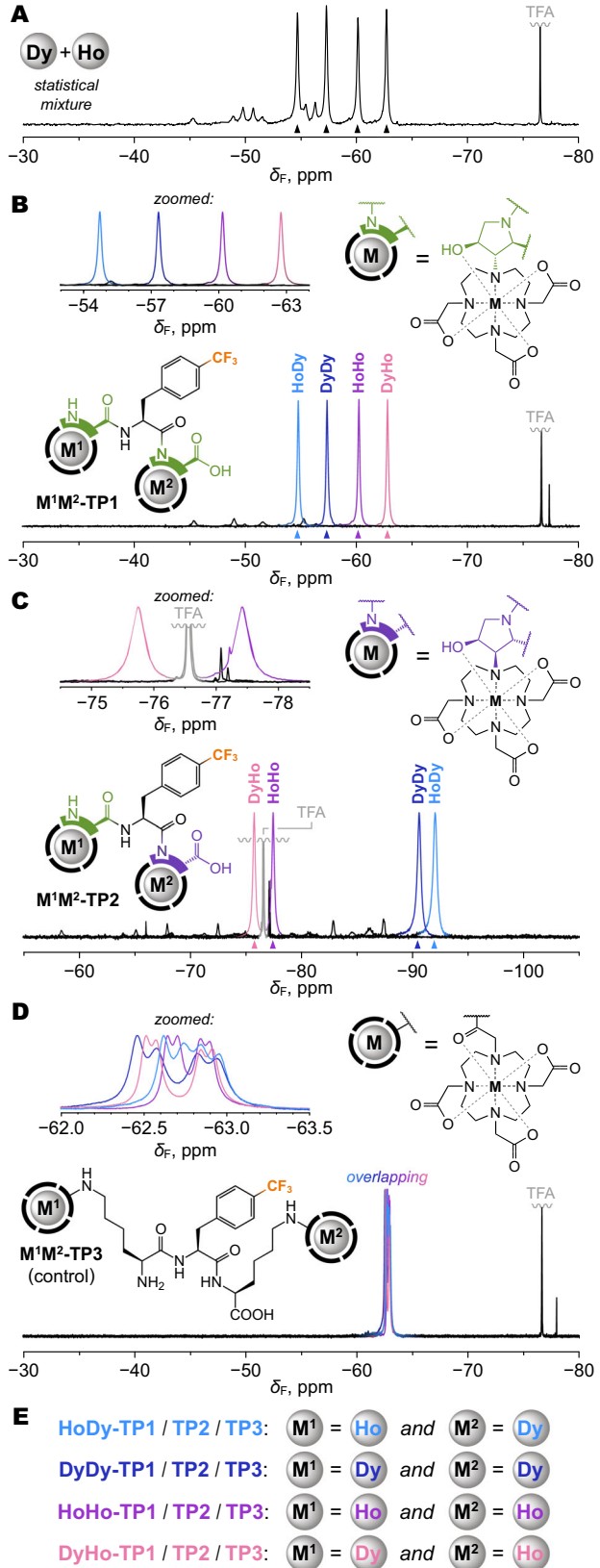

**Fig. 6 $^{19}$F NMR spectroscopy (470.4 MHz, $T = 298.1$ K) of molecules encoded with Dy$^{3+}$/Ho$^{3+}$ ions.** Spectra were measured in aq. MOPS/NaOH buffer (pH = 7) with external D$_2$O for frequency lock. TFA (trifluoroacetic acid) was used as internal NMR reference (−76.55 ppm). Charges of Ln$^{3+}$ ions were omitted for clarity. **A** Statistical mixture resulting from a post-synthetic complexation of Dy$^{3+}$/Ho$^{3+}$ ions provided four major peaks (marked with triangles) with clearly distinguishable shifts representing all M$^1$M$^2$ permutations of the metal ions. The mixture was measured without purification. **B** M$^1$M$^2$-TP1 compounds encoded with specific sequences of Dy$^{3+}$/Ho$^{3+}$ ions, prepared via controlled synthesis, each show one major peak matching with the statistical mixture in the panel above. **C** M$^1$M$^2$-TP2 compounds encoded with the same sequences of Dy$^{3+}$/Ho$^{3+}$ ions, prepared via controlled synthesis, each show one major peak, but the shifts (both absolute and relative) are very different from the analogous M$^1$M$^2$-TP1 compounds in the panels above. **D** M$^1$M$^2$-TP3 control compounds encoded with the same sequences of Dy$^{3+}$/Ho$^{3+}$ ions fail to produce distinguishable peaks. **E** Legend for color-coding of M$^1$M$^2$ sequences.

Alterations to the structure of the building blocks present an opportunity to optimize future molecular designs to achieve better spaced-out NMR signals with fewer overlaps.

**Sequence decoding.** Current strategies to decode sequences of nongenomic oligomers mostly rely on their sequential degradation (e.g., Edman degradation) or fragmentation with tandem MS techniques[8–10], resulting in the destruction of the molecules. Non-destructive spectroscopic methods typically cannot read sequences of monomer units. This is different for paramagnetic metal ions, where the magnetic susceptibility tensors bring directionality. Metal ion sequences in the **M$^1$M$^2$-TP1** system can be decoded from one-dimensional NMR spectra, provided that the $\delta_p^1$, $\delta_p^2$ components are known and the $\delta_F$ shifts are uniquely distinguishable. The latter condition is somewhat limiting, but, nevertheless, suitable sets of elements can be identified. We found by simulation that Tb$^{3+}$, Dy$^{3+}$, Ho$^{3+}$, and Yb$^{3+}$ can be freely combined within the **M$^1$M$^2$-TP1** system, with a minimum peak distance of 0.36 ppm (Supplementary Fig. 17). Moreover, a particularly important property of paramagnetic encoding is that the basic parameters of the system (i.e., $\delta_p^1$, $\delta_p^2$, $\delta_d$ values in the case of **M$^1$M$^2$-TP1**) can be obtained from limited experimental data and then used to reliably simulate and identify sequences that were not yet encountered. This is significant for the design of advanced future systems, where the number of combinations may be too high to be comprehensively explored by experimentation. We tested the reliability of the predictions and sequence decoding on the sub-system of **M$^1$M$^2$-TP1** compounds encoded with Tb$^{3+}$/Dy$^{3+}$/Ho$^{3+}$/Yb$^{3+}$ ions. To ensure that we erased prior knowledge, we excluded from the experimental data all cases where these elements were combined in statistical mixtures. An unbiased set of $\delta_p^1$, $\delta_p^2$, $\delta_d$ values was fitted from this reduced dataset (Supplementary Fig. 18). Using these parameters, we simulated shifts for all 16 permutations. The best match between the experimental $\delta_F$ and simulated $^{sim}\delta_F$ shifts confidently identified all 12 heteronuclear M$^1$M$^2$ sequences (Supplementary Fig. 19), none of which was used to obtain the $\delta_p^1$, $\delta_p^2$, $\delta_d$ values that allowed prediction of the $^{sim}\delta_F$.

thus showing a reasonable agreement with theory (Supplementary Fig. 16), although the accuracy of Bleaney's theoretical model has recently been questioned[35]. Limited synthetic yields prevented us from performing a similar extensive analysis on the **M$^1$M$^2$-TP2** system, but from the comparison in Fig. 6, it is clear that the $\delta_p^1$, $\delta_p^2$ values must be very different from **M$^1$M$^2$-TP1**.

**Multiplexing and combinatorial implications.** Multiplexing is a way to encode information by combining unique signals into composite patterns that can be received through a single channel and decoded back to the original components. Chemical compounds are usable for this purpose by virtue of their signals being detectable by analytical methods. For example,

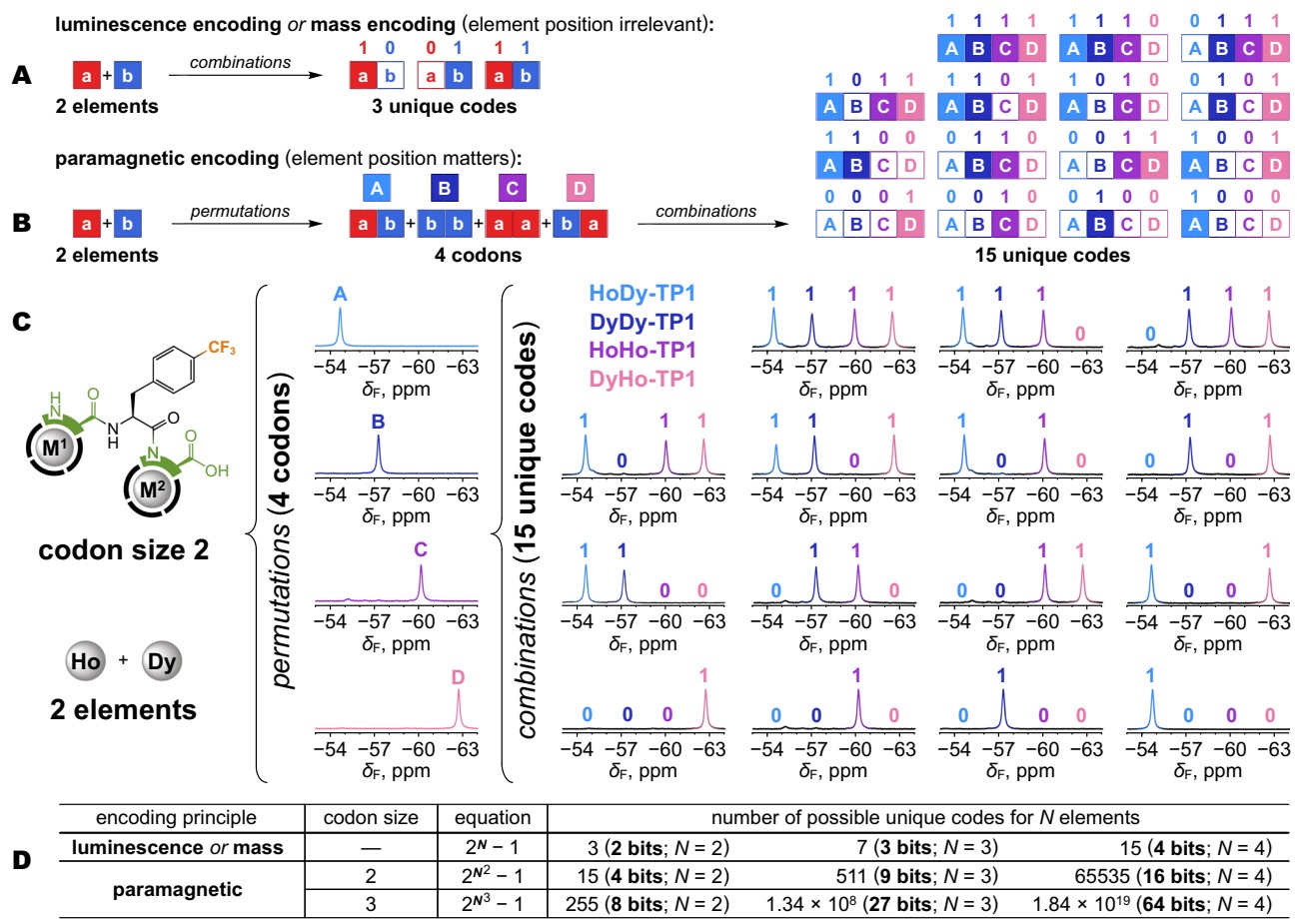

**Fig. 7 Molecular codes achievable with multiplexing. A** Methods that rely on direct detection of the elements (e.g., by luminescence or mass) provide a maximum of three unique codes from two elements. **B** Paramagnetic encoding with two elements provides four permutations (codons) with unique signals that are further multiplexed into 15 unique codes. **C** Physical realization of paramagnetic encoding and multiplexing with $Ho^{3+}/Dy^{3+}$ ions in **$M^1M^2$-TP1** system leading to 15 unique codes. The color-coding of the NMR peaks corresponds to specific $M^1M^2$ permutations as shown in the top middle of this panel. **D** Enumeration of codes achievable with methods of direct element reading vs. paramagnetic encoding for 2–4 elements and considering codon size of two or three elements for paramagnetic encoding. The null case (all zeros) is ignored, hence the –1 term in the equations.

digital files have been encoded into mixtures of organic compounds readable by mass spectrometry[19,20] or fluorescence spectroscopy[23]. Lanthanides were previously multiplexed to generate molecular barcodes based on their unique isotope mass[55], luminescence[24], or paramagnetic NMR shifts[45]. However, decoding of such chemical systems typically proceeds with Boolean logic, considering only the presence/absence of each component, and ignoring their repetition and order. This severely limits the number of codes that can be generated, especially when the number of basic components itself is limited (e.g., chemical elements). For example, two elements can provide only three unique codes readable by luminescence or mass (Fig. 7A). Notably, this applies also to multiplexing NMR signals of molecules that carry a single lanthanide ion[45]. Paramagnetic encoding based on multiple lanthanides within each molecule improves the combinatorics by inserting permutations into the process. First, two elements combine into a codon, where repetitions and order are recognized, resulting in 4 permutations with unique signals. In the second step, these codons are multiplexed, providing 15 unique codes (Fig. 7B). A comprehensive physical realization of this two-level encoding is demonstrated in Fig. 7C with the **$M^1M^2$-TP1** system and $Dy^{3+}/Ho^{3+}$ ions. Controlled synthesis was essential for the whole encoding process to work. If uncontrolled post-synthetic complexation was used instead, the system would revert to only 3 out of the 15 codes displayed in Fig. 7C: 0010 ($Dy^{3+}$), 0100 ($Ho^{3+}$) and 1111 ($Dy^{3+}/Ho^{3+}$ mixture). The combinatorial implications of the two-level encoding are very significant, as the number of unique codes grows with a double-exponential dependency on the number of elements. Paramagnetic encoding thus vastly outperforms encoding based on luminescence, mass, or other methods that read signals directly from the basic components (Fig. 7D). The current **$M^1M^2$-TP1** prototype is capable of at least 16-bit encoding (65,535 codes) with the $Tb^{3+}/Dy^{3+}/Ho^{3+}/Yb^{3+}$ set discussed above that provides 16 distinguishable signals. Theoretically, the combinatorial power can be further improved by increasing the codon size to 3 elements (Fig. 7D). The major limitation is the number of spectroscopically distinguishable codons. However, the NMR spectral window is large in comparison e.g., to optical methods[12,23,24] and can accommodate hundreds of signal peaks at high magnetic fields. Moreover, even overlapping signals may be distinguishable by computational methods such as deconvolution, or thanks to different $T_1$ or $T_2$ relaxation times induced by the paramagnetic ions[53,54]. With an optimized molecular system and method of reading, 64-bit (~$10^{19}$ codes) or higher encoding is conceivable. This would be enough to cover realistic needs for molecular barcodes[1,2,13]. For

**Digital information encoding and reading**. To demonstrate a practical example of information encoding, we employed the system of $M^1M^2$-TP1 and $Tb^{3+}/Dy^{3+}/Ho^{3+}/Yb^{3+}$ ions. All 16 possible compounds were individually synthesized and provided unique and distinguishable NMR signals (Supplementary Fig. 20). With each signal representing one bit (present = 1, absent = 0), it is possible to encode 16 bits of digital data by multiplexing the compounds. Standard ASCII (American Standard Code for Information Interchange) characters use 7 bits, so a single mixture can carry two characters. That is not enough for practical purposes. Therefore, we organized five different mixtures within a capillary into layers that were spatially separated by an immiscible signal-free solvent $CCl_4$ (Fig. 8A). Thus, a composite NMR sample was created with the capacity to store ten ASCII characters, enough to encode a reasonably strong password. To read back the information, we used $z$-resolved NMR spectroscopy to acquire spectra of all layers simultaneously (Fig. 8B). After referencing the chemical shifts in each layer to an internal standard (trifluoroacetic acid, TFA), the presence/absence of the signals was evaluated (Fig. 8C) and converted to a binary code (Fig. 8D). Due to the lower spectral resolution of $z$-resolved NMR, we avoided the confusion of the two least-resolved peaks (DyDy/YbTb, Fig. 8C) by shifting the beginning of the binary code to one of them (DyDy), thus eliminating it by definition (binary codes of all standard ASCII characters start with 0, Fig. 8E). Conversion of the binary code to characters revealed the encoded 10-character password (Fig. 8F, G). It is important to note that multiple pairs of positional isomers (e.g., YbHo/HoYb) present in the mixtures make the encoded information practically unreadable with methods other than magnetic resonance since the isomers are very difficult to distinguish by other means. This is a very attractive feature especially for potential counterfeit applications of such molecular codes.

**Parallel reading with MRI**. A key aspect of reading and communicating information is whether it proceeds in a sequential or parallel way. Parallel systems offer higher and more scalable throughput, which is also suitable for imaging applications. For example, optical reading methods capture information simultaneously from many objects in the field of view, while MS can only process one sample at a time. Magnetic resonance can provide both spectroscopic and spatial information simultaneously, thus allowing parallel reading and imaging. Spectrally resolved MRI has been previously realized with various systems based on chemically different diamagnetic compounds[38–41], host-guest interactions[42,45], and lanthanide-induced NMR shifts[43–45]. The latter approach offers the advantage to select the NMR shifts without changing the chemical properties, which is particularly useful for in-vivo imaging applications[43,44]. However, previous systems based on a single lanthanide per molecule offered only a very limited choice. The programmable $M^1M^2$-TP1 system provides an advanced level of control to generate a higher number of practically usable NMR signals. To demonstrate this, we used just two lanthanide ions $Dy^{3+}/Ho^{3+}$ to encode the $M^1M^2$-TP1 molecules, which were then pipetted into a $7 \times 5$-well plate in patterns to write one letter with each compound. Each well thus contained a homogenous mixture of 0–4 compounds (Fig. 9A). The whole well plate was imaged on a preclinical 4.7 T MRI scanner with a $^{19}F$ CSI (Chemical Shift Imaging) pulse sequence, where two dimensions represent space coordinates and the third is a frequency domain. The information contained in the 35 wells

was obtained simultaneously, the encoded molecules were distinguished owing to their different resonance frequencies (Fig. 9B), and the four letters could be resolved and independently displayed (Fig. 9C–F). A fly-through view of the MRI data is provided in Supplementary Movie 1.

**Limit of detection**. NMR spectroscopy is typically not considered as a method for reading molecular codes because of its relatively low sensitivity compared to e.g., mass spectrometry. This prompted us to investigate the limit of detection achievable with $M^1M^2$-TP1 molecules encoded with $Dy^{3+}/Ho^{3+}$ ions. Previous works have demonstrated that shortening of longitudinal relaxation times ($T_1$) induced by paramagnetic ions allows to use a rapid scan rate and boost sensitivity by acquiring more signal per time[53,54]. We took advantage of this effect in $M^1M^2$-TP1 compounds (Supplementary Fig. 21). The signal-to-noise ratio (SNR) in NMR grows with the square root of the number of scans, but there are practical limits to the experiment time. We extrapolated the detection limit at 11.7 T and SNR = 3 to be 5.8 nmol with a short measurement time (1.6 s, 32 scans), or 123 pmol with a reasonably long measurement time (60 min, 72,000 scans). Paramagnetic codes are therefore conceivably usable as identification tags in bead-based bioassays (a single 90-μm TentaGel bead can hold 100 pmol). The detection limit could be further lowered by technical as well as chemical means, e.g., by optimization of the NMR hardware and pulse sequence, optimizing the molecular system to provide narrower NMR signals (higher SNR), or including a higher number of magnetically equivalent $^{19}F$ nuclei. Although the conventional NMR/MRI technology is large and expensive, in principle it can be re-designed for the purpose of reading paramagnetic molecular codes. Downsizing using cost-efficient permanent magnets can dramatically lower the cost and potentially improve the sensitivity to match a particular application[56,57].

## Discussion

Properties of paramagnetically encoded molecules depart from typical chemical systems and approach macroscopic hardware and software. As the encoded information is hidden in chemically inaccessible f-electrons of lanthanides, the hardware (molecule) remains physically unchanged (isostructural) when loaded with different software (metal ion sequence). Simple and clear programming rules, which can be calibrated from a small number of synthesized examples, provide predictable behavior of the molecular code. Reading in the RF spectrum draws parallels with the technology of radiofrequency identification (RFID) tags by being repeatable and non-destructive, and by working through space without a direct line of sight. The paramagnetic codes presented here are nearly chemically identical, based on elements with negligible biological background (lanthanides, fluorine), and structurally related to clinically approved MRI contrast agents. These are favorable qualities for tagging biological objects or artificial objects in the biological milieu. We refer to the $M^1M^2$-TP1 and $M^1M^2$-TP2 systems as prototypes, because many aspects of their modular construction can be further optimized to build more advanced systems with enhanced functionality. The major roadblock to immediate practical use is the low sensitivity of reading with conventional NMR technology. However, rapid advancements in quantum sensors may soon make the detection of single molecules a routine reality[58,59], opening the door to as-yet unforeseen applications.

## Methods

**Synthesis of building blocks, tripeptides, and their metal chelates**. Detailed synthetic strategies and procedures, including the conditions for the separation of isomers and characterization of all products, are provided in Supplementary Information. All syntheses proceeded in solution. Some lanthanide ions were excluded

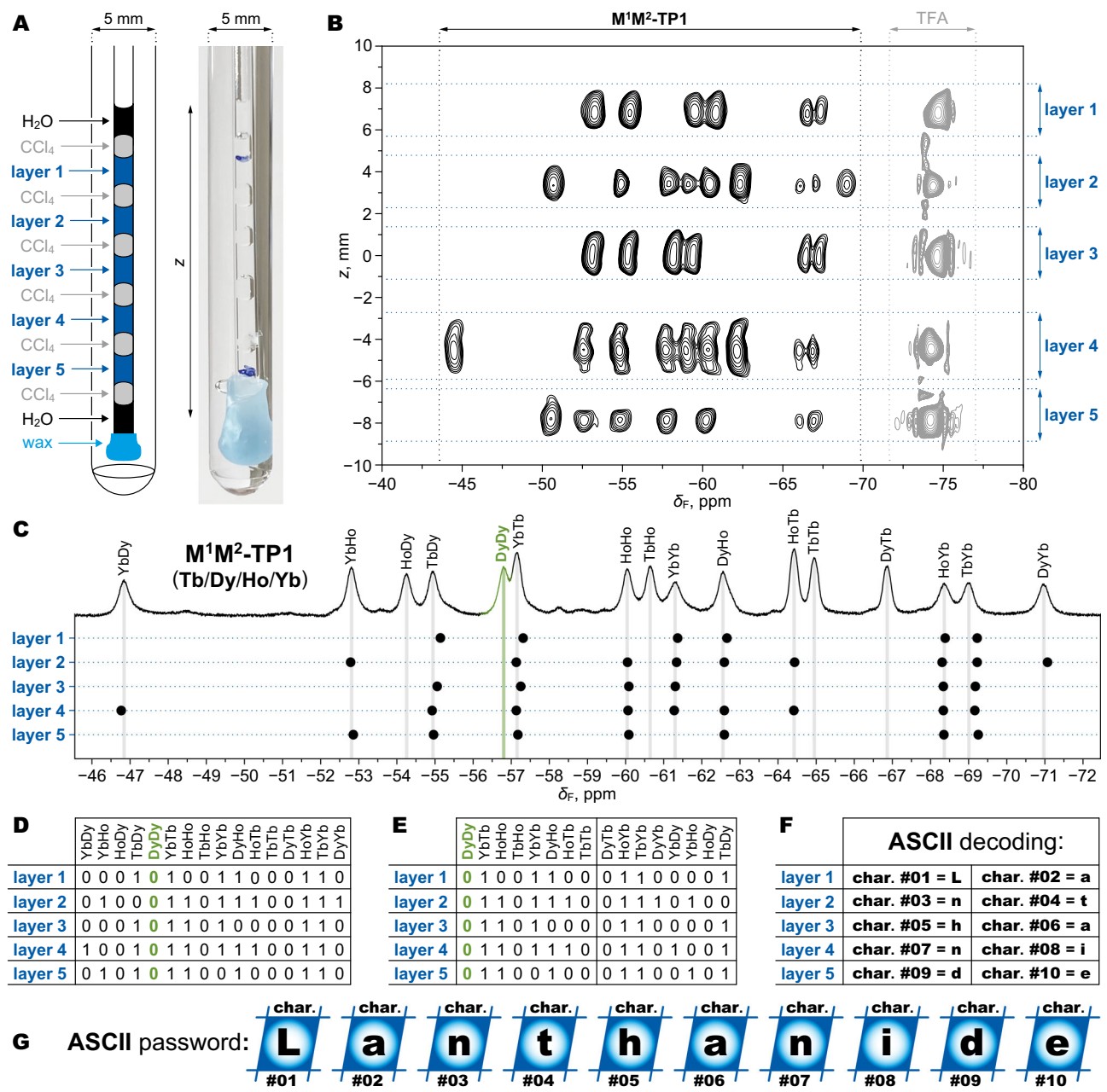

**Fig. 8 Password decoding from a single sample with z-resolved NMR. A** Physical realization of a single NMR sample containing 5 different mixtures of **M¹M²-TP1** compounds encoded with $Tb^{3+}/Dy^{3+}/Ho^{3+}/Yb^{3+}$ ions. Aqueous solutions (in MOPS/NaOH buffer, pH = 7) of the mixtures (layers 1–5) are separated by layers of $CCl_4$. The capillary is submerged in $D_2O$. **B** Z-resolved $^{19}F$ NMR spectrum (470 MHz, $T = 298.1$ K) of the sample from panel A shows signals of the **M¹M²-TP1** compounds and internal standard (TFA, trifluoroacetic acid) resolved for the layers along the z-axis. Shifts from different layers are not perfectly aligned due to bulk magnetic susceptibility (BMS) effects that varied for each layer. **C** Evaluation of the signals from panel B by comparison with an independently obtained 1D NMR spectrum of a mixture containing all 16 **M¹M²-TP1** compounds. Note that peaks of DyDy and YbTb are close but distinguished in the 1D spectrum. Black dots mark shifts from the z-resolved spectrum after correcting for BMS effects in each layer (referencing to TFA). Good match is achieved between expected shifts (vertical lines) and determined shifts (black dots). **D** Binary representation of the information from C. **E** Final table of the binary codes for 10 ASCII (American Standard Code for Information Interchange) characters encoded in the sample. Relative to D, the binary codes are rearranged to start at the position of DyDy (green shading), which is not used to avoid confusion with YbTb. **F** ASCII characters obtained by conversion from the binary codes in E with indication of their order in a password. **G** The final decoded 10-character password.

from testing for various reasons ($La^{3+}$: redundant diamagnetic, $Ce^{3+}$: 3 + /4+ redox chemistry, $Pm^{3+}$: radioactive, $Gd^{3+}$: inducing no paramagnetic shift). Concentrations of $Ln^{3+}$ ions in stock solutions for synthesis and in solutions of **M¹M²-TP1**, **M¹M²-TP2**, and **M¹M²-TP3** compounds were determined with ICP-AES.

**X-ray crystallography**. Detailed procedures for preparation of the single crystals of $[Dy(L^1)]·3.5H_2O$ and $[Dy(L^2)]·3H_2O$, details of structure solving and

refinement, as well as additional details of the obtained structures are provided in Supplementary Information (Supplementary Methods and Supplementary Figs. 1–3, 5–7, 65, 97).

**$^{19}F$ NMR spectroscopy**. Measurements of all $^{19}F$ NMR spectra were done on Avance III$^{TM}$ HD 500 MHz spectrometer (*Bruker*, 470.4 MHz for $^{19}F$) equipped with a broad-band cryo-probe with ATM module (5 mm CPBBO BB-$^1H/^{19}F/^{15}N/D$ Z-

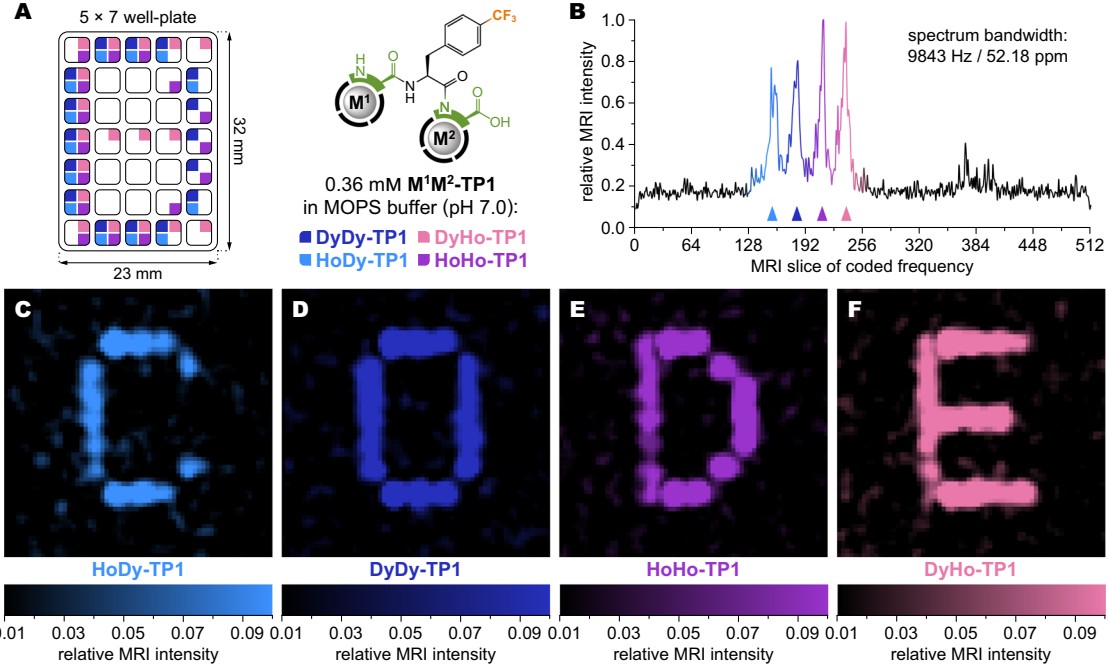

**Fig. 9 Image encoding and reading with ¹⁹F MRI (4.7 T). A** Distribution pattern of **M¹M²-TP1** compounds encoded with $Dy^{3+}/Ho^{3+}$ ions within a 7 × 5-well plate (final concentrations ~0.36 mM each compound, well volumes completed to 75 µL with water). **B** ¹⁹F MR spectrum obtained from the entire well plate shows four distinct peaks of the four encoded compounds (marked with triangles, color-coding as explained in A). The spectrum was created by projecting the maximum signal intensity from each slice of (CSI) MRI data onto the frequency dimension. The spectral resolution was 19.2 Hz (0.10 ppm) per slice. **C–F** Selective display of **HoDy-TP1** (**C**), **DyDy-TP1** (**D**), **HoHo-TP1** (**E**) and **DyHo-TP1** (**F**) compounds reveals the patterns encoded with them into the well plate as legible letters (average of 15 adjacent slices for each compound peak). Displayed field of view: 50×50 mm, voxel size 0.78 × 0.78 × 10 mm. Image intensities were mapped to colors as shown in the legends. See also Supplementary Movie 1 for a fly-through view of the MRI data.

GRD). Solutions of samples were prepared in MOPS/NaOH buffer (pH = 7) and placed into an insert capillary that was inserted into a 5-mm NMR tube containing $D_2O$ (used for NMR frequency lock). The volume of solution in the insert observable by the spectrometer probe was ~50 µL. Longitudinal relaxation times ($T_1$) were obtained by inversion recovery sequence. Spectra of statistical mixtures or individually synthesized **M¹M²-TP1** and **M¹M²-TP2** compounds were measured with these parameters: 90° observation pulse, pre-acquisition delay = 1 s, acquisition time = 0.58 s and 512 or 4096 scans. Measurements of SNR were performed on a sample containing a mixture of **HoDy-TP1**, **DyDy-TP1**, **HoHo-TP1**, and **DyHo-TP1** compounds, each at molar amount of 0.18 µmol in the volume observable by the NMR probe (50 µL). Parameters for SNR measurements were: pre-acquisition delay = 0, acquisition time = 50 ms, receiver gain maximum value = 203. The SNR values were obtained using *MestReNova* software (version 12.0.3) after the application of an exponential window function of 1 Hz. The limit of detection (SNR = 3) was extrapolated from SNR measured with 32 scans (1.6 s total time), considering that SNR scales linearly with the analyte concentration and with the square root of the number of scans. Further details for SNR measurements are provided in Supplementary Fig. 21. ¹⁹F $z$-resolved NMR spectrum was acquired using a phase encoding pulse sequence with resolution 2048 × 64 points at 20 mm field of view, number of scans = 32 K, acquisition time = 18 ms, relaxation delay = 100 µs (for technical reasons). The excitation pulse was ~31°, which was the result of optimization aiming to enhance the SNR of weaker (broader) signals, sacrificing the SNR of sharper peaks. This was possible due to the differences in $T_1$ relaxation times. The total experimental time was ~12 h. For subsequent processing, only 700 × 64 raw points were used in order to reduce the noise. Data were apodized using QSINE window functions in both F2 (SSB = 10) and F1 (SSB = 1) dimensions and zero-filled to 8192 × 128 points. Processing was done in *Bruker TopSpin 3.5* software. Spectra were visualized using the *NumPy* and *Matplotlib Python* libraries.

**Magnetic resonance imaging.** Stock solutions of individually synthesized **HoDy-TP1**, **DyDy-TP1**, **HoHo-TP1**, and **DyHo-TP1** compounds in 0.5 M MOPS/NaOH buffer (pH = 7.0) were pipetted into a 7 × 5-well plate (cut out of standard 384-well plate) and the volume in each well was completed with water to 75 µL to reach compound concentrations ~0.36 mM each. The well plate was covered with plastic tape and imaged on 4.7 T scanner Bruker Biospec 47/20 (*Bruker BioSpin*, Ettlingen, Germany) with a custom-built dual ¹H/¹⁹F RF surface coil. First, ¹H MRI was performed in all three planes (axial, coronal, and sagittal) for localization of the sample. Then, ¹⁹F NMR spectra were acquired by 90° single pulse sequence to precisely determine resonance frequencies of the

compounds. ¹⁹F MRI images were obtained using a CSI sequence (120,000 scans, repetition time = 700 ms, field of view = 50 × 50 mm; slice thickness = 10 mm, matrix of acquired image = 16 × 16 × 512, matrix of reconstructed image = 64 × 64 × 512, acquisition time = 23 h 20 min, resonance frequency = 188,630,130 Hz, bandwidth = 9843 Hz/52.18 ppm). Images for display were prepared in *Matlab* software. First, the data in the Bruker format were imported to *Matlab* using read_2dseq function (by Cecil Yen (2021), https://www.mathworks.com/matlabcentral/fileexchange/69177-read_2dseq-quickly-reads-bruker-s-2dseq-mri-images). Maximum intensity projection along the frequency axis was used to find the peaks of the respective samples. Background signal (outside of main peaks) was calculated as an average of 41 slices (# 10–50 perpendicular to the frequency axis) and subtracted from each slice of the original data. Then, the intensity was normalized to 1 for the highest intensity. The final images displayed in Fig. 9 were obtained by averaging 15 slices (perpendicular to the frequency axis) around the maximum for each peak, and mapping the image intensity to specific color maps. Supplementary Movie 1 was created analogously, with each frame showing a floating average of 15 slices.

**Data fitting.** Fitting of experimental shifts $\delta_F$ from statistical mixtures of **M¹M²-TP1** compounds to obtain $\delta_p^1$, $\delta_p^2$, $\delta_d$ contributions was done according to Eq. (1) with the method of least squares using the *Solver* function in *Microsoft Excel*. Sequences of $Dy^{3+}/Ho^{3+}$ synthesized as individual compounds were the starting point for the assignment of $\delta_F$ to particular $M^1M^2$ sequences. Next, homonuclear sequences were identified based on their repeated occurrences between mixtures containing the particular element. The assignment of heteronuclear sequences then proceeded simultaneously with the fitting, one statistical mixture at a time. If a satisfactory fit could not be obtained, it indicated that the assignment was wrong and the positions in $M^1M^2$ sequences were exchanged. This was continued until consensus was reached across all data in a final global fit. The paramagnetic shift contributions of the $Sm^{3+}$ ion were too small to be obtained simultaneously with the other lanthanide ions. Therefore, $\delta_p^1$, $\delta_p^2$ parameters for $Sm^{3+}$ were fitted separately from the $Sm^{3+}/Y^{3+}$ statistical mixture of **M¹M²-TP1** compounds and were then used as fixed parameters in the global fit.

## Data availability

The crystallographic data for the structures reported in this study have been deposited at the Cambridge Crystallographic Data Centre (CCDC) under deposition number 2072548 for [**Dy(L¹)**]·3.5H₂O and 2143481 for [**Dy(L²)**]·3H₂O and can be obtained free of charge

from the Centre via its website (www.ccdc.cam.ac.uk/getstructures). All other data supporting the findings in this study are available within the article and its Supplementary Information, as well as from the corresponding author upon request.

## Code availability

No custom computer code was used in this study.

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

## Acknowledgements

This work was supported by the Czech Science Foundation (17-22834Y, M.P.). Additional support is acknowledged from the Student Grant Scheme at the Technical University of Liberec (SGS-2022-3002, M.V.), and from the Ministry of Health of the Czech Republic - conceptual development of research organization (Institute for Clinical and Experimental Medicine–IKEM, IN 00023001, D.J.). The authors thank to Kelsea Grace Jones for language editing, Veronika Urbanová for proof-reading and Adam Jaroš for molecular modelling.

## Author contributions

M.P. conceived the project and supervised the work. J.K. synthesized the compounds. R.J. and M.K. designed and conducted the synthesis of the epoxide precursor. M.D. and O.S. conducted NMR measurements. D.J. and M.V. conducted MRI experiments and constructed a radiofrequency coil. I.C. performed X-ray diffraction and solved the structures. T.D. created the artwork. M.P., J.K., and T.D. wrote the manuscript. All authors proofread and approved the manuscript.

## Competing interests

M.P. and J.K. are co-inventors on a pending patent application no. PCT/CZ2020/050032 filed in the name of applicant Ustav Organicke Chemie a Biochemie AV CR V.V.I. The application covers some of the compounds discussed in this work. The remaining authors declare no competing interests.
