## [Peer Review File · Nature Communications]

Paramagnetic encoding of moleculesREVIEWER COMMENTS

Reviewer #1 (Remarks to the Author):

The manuscript "Paramagnetic encoding of molecules" describes the design of paramagnetic molecular scaffolds that provide sequence-specific characteristics of magnetic susceptibility tensors. Each of the obtained lanthanide-sequences introduced into the synthesized rigid molecular architecture provides a different ^{19}F -NMR shift offset of the peak of the trifluoromethyl functional group of the same scaffold. Interestingly, the authors show that the sequence in which the paramagnetic lanthanides are ordered results in a combined magnetic susceptibility tensor, thus increasing the number of permutations that could be reached with their design. Applying the approach to ^{19}F -chemical shift imaging (CSI), the authors demonstrate the potential to spatially decode the information encoded by the paramagnetic molecules. Overall, this is a very interesting approach, which combines magnetic susceptibility tensors of lanthanide ions within a single molecule, thus enriching the number of possible chemical shifts when using a set of given lanthanides. Therefore, the approach might have multiple implications in information encoding in the future, but as it relies on ^{19}F -NMR and ^{19}F -MRI, I am very skeptical that it will be applicable to any sort of system for programmable digital molecular information. The synthetic work is impressive and the data support the hypothesis. However, having shown, yet another, molecular system with multiplexed MR capabilities but without demonstrating any specific example for information encoding/decoding or clearly convincing of the advantages of the proposed approach over other (many, not cited) multiplexed MRI systems, I don't think this work shows the originality required for publication in Nature Communications.

Specific comments:

1. I am very discouraged by the fact that the authors chose not to refer to previous relevant literature regarding other approaches through which multispectral/multicolor/multiplexed MRI is demonstrated. There are many examples, either more-than-a-decade old or very recent ones, of multiplexed MRI based on other molecular systems, ranging from paramagnetic agents, ^{19}F -based formulations, micro-magnets, CEST or combinations thereof (paraCEST, para- ^{19}F and ^{19}F -CEST). These works should not be just cited but also added to the introduction in order to allow readers not familiar with the field to put this work into the right context. The way the introduction is shaped now is somehow misleading, reflecting the advantage of the work over MS-based approaches without mentioning other solutions based on magnetic resonance, which have been continuously proposed over the last 15 years.
2. In this regard, the authors should specifically discuss the advantages and shortcomings of their paramagnetic encoding approach over other strategies. Beyond proposing another way of spectral encoding by MRI, they should explain what the current approach offers that other MR-based approaches cannot.
3. Some examples of fluorescent-based colors used for molecular encoding should be given in the introduction as these types of molecules are still the most common ones for molecular encoding/decoding.
4. It would be very beneficial if the authors show a concrete demonstration in an MRI setup of the spatial multiplexing capabilities of the example shown in Figure 8. It looks like the paramagnetic encoding here was achieved by multiple 1D NMR acquisitions of numerous (and different) samples. A 2D spatial display of the information encoded by multiple probes of the

same sample is important for the method's establishment. Would it possible to map multiple molecular probes that have different combinations of Ln at the same imaging voxel and differentiate between them (i.e., decode the information encoded by them)?

5. The authors use nonscientific subjective/biased/not-conclusive wording (especially in the introduction) that should be avoided. Some examples are "all areas" in line 27, "must be" in line 28, "absolutely crucial" in line 42, "existence of life itself" in line 43 and "ultimate frontier" in line 45.

6. Lines 46-47 of the introduction describe DNA-based sensors for encoding digital data. Nevertheless, other examples based on different types of molecules should be given.

7. Does the chirality of DO3A-Hyp affect the obtained results? Would you get different magnetic susceptibility tensors when using the R-Hyp isomer rather than a S-Hyp one? Would the chirality affect the PCS of a single lanthanide? An example would be very beneficial and a discussion of this issue is missing.

8. The authors mention that one of the advantages of their approach (and maybe of other MRI-based approaches) over MS-based strategies is the ability to recycle the probe. Would it be possible to recycle it? Would one be able to separate between different probes located in a single mixture?

9. In theory, there are many more combinations possible compared to the ones that can be practically used. The authors should discuss issues of spectral resolution and how this parameter affects the capability of the molecular system proposed.

10. Is an 8-bit code indeed applicable? For example, DyDy shows a chemical shift of -57.40 ppm, and TmTm of -57.61 (Table Supplementary Figure 9). Or, perhaps, this table is just not properly explained.

Reviewer #2 (Remarks to the Author):

The article reports the design, synthesis, and characterization of a bimetallic system that can act as a molecular versatile platform for information encoding, readable by MRI. While the size of current information storage devices is reaching physical limitations, the concept of encoding and decoding information at the molecular level has never been more relevant. The presented manuscript proposes an elegant approach to this challenge, well supported by a sound and well-explained methodology.

Unfortunately, the principle, described here as unprecedented, is very similar to a very recent study published in *Nature Communication* (*Versatile non-luminescent color palette based on guest exchange dynamics in paramagnetic cavitands*) by Amnon Bar-Shir and co-workers. doi: 10.1038/s41467-021-23179-9).

Following are my suggestions and comments on the presented manuscript:

- The introduction would benefit from a general scheme explaining the principles of the system and approach reported in this manuscript.
- The use of the term "monomer building block" is ambiguous as to what precise molecule it refers to. I initially understood the term for the Dy complex (monomer and dimers usually used for the number of metallic centers in the molecules). If I am not mistaking, it refers to the ligand L^{1-} that is used as a synthon in the different synthetic approaches. The manuscript

would benefit from further precision at the beginning of the paragraph.

- Page 7, “¹⁹F NMR spectra showed four major singlets”. Do you have a hypothesis on why there are more signals present? On the Supp. Fig. 3, are all integrations equal to 1 (especially for Nd/Tb)?
- Please amend the first sentence of the ‘Rules of paramagnetic encoding’ paragraph in consequence of the recently published paper mentioned above.
- ‘Sequence decoding’ paragraph: I found it hard to follow the reason why a bias-free set of parameters needed to be obtained. I would suggest to the authors the addition of a sentence or two before the introduction of this part on why this is necessary for future applications. Although stated at the end of the paragraph, it would help with the flow of the manuscript.

Minor Details:

- Main text page 22, I am not convinced by the formulation of the complex as the asymmetric unit formula $[\text{Dy}(\text{L})]_2 \cdot 7\text{H}_2\text{O}$ as it implies a dimerization, which is not the case here. I suggest describing either the repeating unit $[\text{Dy}(\text{L})]_2 \cdot 3.5\text{H}_2\text{O}$ or marking the “polymerization” as $[\text{Dy}(\text{L})]_n \cdot 3.5\text{H}_2\text{O}$ (with a personal preference for the second notation).
- ESI page 5, ‘*direct methods*’ should be replaced by ‘*intrinsic phasing*’, which is ShelXT (direct methods would be ShelXS).
- ESI page 5 and Main text page 23: Add the unit Å to the Mo wavelength.
- ESI page 7, ‘*D. Rearrangement*’ should read ‘*D. Rearrangement*’

X-Ray Diffraction study:

After careful examination of the crystal structure obtained and the chemicals involved in the synthesis, I do not have a better interpretation than the one proposed by the authors regarding the high residual density Q-peak located near the macrocycles. Therefore, I recommend the publication of the CIF file as presented.

However, the description of the rather complicated structure would benefit from a slight reorganization. I always find it clearer when the structure is described “increasing the scale” with corresponding figures associated with each step: I would suggest starting with the description of the mono-nuclear complex on its own, then the asymmetric unit where two complexes are present, followed by the helix 3D arrangement, then packing of the two helices. Maybe the term “repeating unit” could be used from the polymer lexicon? Figures could be made clearer by using wireframe representation of the carbon atoms (except in the first description of the complex where ellipsoids are necessary).

Reviewer #3 (Remarks to the Author):

This is a very interesting and well-executed work that presents a novel and exciting concept: The use of paramagnetic lanthanides to encode information at the molecular level. For this purpose the authors designed a ligand scaffold containing an amino acid moiety that can be used to join together two different paramagnetic lanthanide ions and a reporter unit containing a

CF3 group. The topic is introduced in a very clear way that non-specialists can follow. The manuscript contains a solid piece of experimental work, including the synthesis of the ligands, NMR spectroscopy and imaging and the characterization of the complexes using X-ray measurements.

For these reasons I support publication of this paper in Nat. Commun. after addressing a few minor issues:

-Page 4: "This tensor has radial and angular dependencies specific for each element". It is not actually only the nature of the element (lanthanide), but the identity of the system as a whole (the complex) that dictates the magnetic anisotropy, and thus the pseudocontact shift. Please rephrase.

-Page 7: "The chemical shift induced by paramagnetic lanthanide ions cannot be predicted from theory with confidence". I agree with the authors that this is not a trivial task, but quite some progress was made over the last few years towards this direction. In particular wave function calculations were shown to predict reasonably well the magnetic anisotropies, and thus pseudocontact shifts, of lanthanide complexes (i. e. Yb(III) derivatives).

- Rules of paramagnetic encoding: The analysis of the paramagnetic shifts presented in the paper is very interesting. The ^{19}F nucleus is rather far away from the paramagnetic center in terms of number of bonds, and thus contact contributions are expected to be negligible. I miss here some comment on the origin of the paramagnetic shifts and their comparison with the predictions made by Bleaney's theory. This theory has been recently put into question by different authors, but still represents an useful guide to design lanthanide paramagnetic systems. I have plotted the paramagnetic contributions shown at the bottom of Supplementary Fig. 7 versus the Bleaney constants. One observes a rather good linear correlation, except for a couple of points that fall off the linear trend. The qualitative trend observed here is nevertheless well reproduced by Bleaney's theory. I understand that the aim of this paper is not to provide a detailed theoretical analysis, but still some comment along these lines is welcome, as it would provide some rational (and confidence) to the analysis.

- The authors should at least provide some comment on the origin of the minor signals present in the NMR spectra (Fig. 5). If avoiding different isomers in solution was a key objective, DO3A derivatives may not be the best choice. I admit however that the population in solution is large dominated by a major isomer, but this is something difficult to predict before synthesizing and characterizing the complexes.

Point-by-point response to the REVIEWER COMMENTS

for

revised manuscript "Paramagnetic encoding of molecules"

Reviewer #1 (Remarks to the Author):

The manuscript "Paramagnetic encoding of molecules" describes the design of paramagnetic molecular scaffolds that provide sequence-specific characteristics of magnetic susceptibility tensors. Each of the obtained lanthanide-sequences introduced into the synthesized rigid molecular architecture provides a different ¹⁹F-NMR shift offset of the peak of the trifluoromethyl functional group of the same scaffold. Interestingly, the authors show that the sequence in which the paramagnetic lanthanides are ordered results in a combined magnetic susceptibility tensor, thus increasing the number of permutations that could be reached with their design. Applying the approach to ¹⁹F-chemical shift imaging (CSI), the authors demonstrate the potential to spatially decode the information encoded by the paramagnetic molecules. Overall, this is a very interesting approach, which combines magnetic susceptibility tensors of lanthanide ions within a single molecule, thus enriching the number of possible chemical shifts when using a set of given lanthanides. Therefore, the approach might have multiple implications in information encoding in the future, but as it relies on ¹⁹F-NMR and ¹⁹F-MRI, I am very skeptical that it will be applicable to any sort of system for programmable digital molecular information. The synthetic work is impressive and the data support the hypothesis. However, having shown, yet another, molecular system with multiplexed MR capabilities but without demonstrating any specific example for information encoding/decoding or clearly convincing of the advantages of the proposed approach over other (many, not cited) multiplexed MRI systems, I don't think this work shows the originality required for publication in Nature Communications.

Specific comments:

1. I am very discouraged by the fact that the authors chose not to refer to previous relevant literature regarding other approaches through which multispectral/multicolor/multiplexed MRI is demonstrated. There are many examples, either more-than-a-decade old or very recent ones, of multiplexed MRI based on other molecular systems, ranging from paramagnetic agents, ¹⁹F-based formulations, micro-magnets, CEST or combinations thereof (paraCEST, para-¹⁹F and ¹⁹F-CEST). These works should not be just cited but also added to the introduction in order to allow readers not familiar with the field to put this work into the right context. The way the introduction is shaped now is somehow misleading, reflecting the advantage of the work over MS-based approaches without mentioning other solutions based on magnetic resonance, which have been continuously proposed over the last 15 years.

Reviewer #1 perhaps places too much weight on the MRI aspect of our work, which is not the central topic. Spectrally resolved MRI has indeed been demonstrated with various chemical systems before. The intention of our work was not necessarily to provide a new type of MRI probes, but rather to demonstrate a new principle of encoding and explore its potential. While we show the possibility of reading the encoded information with MRI, we place more emphasis on reading with NMR spectroscopy/NMR spectrometer to demonstrate the novelty and possibilities of paramagnetic encoding.

However, per Reviewer's request, we now make multiple references to previous works where various chemical systems, with or without lanthanide ions, were used for multiplexed MRI:

- (1) Page 6, line 111-115 (Introduction).
- (2) Page 11, line 205 - 206 ("Rules of paramagnetic encoding").

(3) Page 16, line 328 - 332 ("Parallel reading with MRI").

We also stress the difference between the previous molecular systems that used a single lanthanide ion per molecule, and the **M¹M²-TP1** system presented in this work. The novelty of our approach lies in combining the effects of two lanthanides within one molecule, which provides better control over the NMR shifts and much higher number of distinguishable NMR signals.

Altogether, 21 new references were added to this revision.

In the introduction, the emphasis placed on the comparison with MS-based methods of reading was deliberate, because MS is probably the only general-purpose technique capable of decoding sequences of monomers from oligomeric/polymeric chains and/or their mixtures. In this sense, MS is superior to NMR (or MRI), which cannot normally read sequences. We wanted to compare our approach with the best available alternative methods of reading.

2. In this regard, the authors should specifically discuss the advantages and shortcomings of their paramagnetic encoding approach over other strategies. Beyond proposing another way of spectral encoding by MRI, they should explain what the current approach offers that other MR-based approaches cannot.

As explained above, the central topic is not spectrally encoded MRI, but more generally to show that combining several paramagnetic lanthanide ions in a controlled way within one molecule provides a new way to encode information into molecules and their mixtures. Namely, our approach provides much higher number of signals (and codes) from a limited number of basic constituents, compared to previous approaches. Moreover, the rules of how the signals arise from contributions of the individual lanthanide ions are predictable and can be "calibrated" from a limited number of examples.

In line with this Reviewer's comment, we expanded the explanations on several places in the manuscript. Regarding the **advantages**:

(1) Page 6, lines 111-117 and 122-123 (Introduction).

(2) Page 13, lines 248-256 ("Sequence decoding"). Here we explain more clearly the advantages of having predictable rules and the possibility to "calibrate" the system parameters from limited data.

(3) Pages 13-14, the whole section "Multiplexing and combinatorial implications". Here we explain how dramatically the number of possibilities grows when two or more lanthanide ions are combined within one molecule.

(4) Page 16, lines 317-321 (newly added section "Digital information encoding and reading"). Here we explain that the existence of positional isomers, such as HoYb-TP1/YbHo-TP1, is a quite unique and potentially advantageous possibility to make the molecular code readable only by magnetic resonance and not by other methods.

(5) Page 16, lines 328-335 ("Parallel reading with MRI"). Here we elaborate on the limitations of the previously reported systems for multiplexed MRI and the advantages that our approach provides (higher number of signals, programmable shifts).

Regarding the **shortcomings**:

(6) Pages 14-15, lines 290-296 ("Multiplexing and combinatorial implications"). Here we discuss the limitations arising from spectral overlaps and potential ways to overcome them.

(7) Pages 17-18, the whole section "Limit of detection". Here we discuss the limitations that regard relatively poor sensitivity of NMR and MRI reading.

3. Some examples of fluorescent-based colors used for molecular encoding should be given in the introduction as these types of molecules are still the most common ones for molecular

encoding/decoding.

Page 5, lines 88-99. We expanded in the introduction the paragraph regarding "Reading" to include approaches based on fluorescent labels and luminescent particles. New references are provided.

4. It would be very beneficial if the authors show a concrete demonstration in an MRI setup of the spatial multiplexing capabilities of the example shown in Figure 8. It looks like the paramagnetic encoding here was achieved by multiple 1D NMR acquisitions of numerous (and different) samples. A 2D spatial display of the information encoded by multiple probes of the same sample is important for the method's establishment. Would it possible to map multiple molecular probes that have different combinations of Ln at the same imaging voxel and differentiate between them (i.e., decode the information encoded by them)?

This suggestion was very useful, and we conducted an additional MRI experiment. Yes, the experiment proved that multiple probes encoded with different lanthanide permutations can be independently mapped within a 2D grid, even if they are present simultaneously in the same imaging voxel. Please see the updated section "Parallel reading with MRI" and Fig. 9.

5. The authors use nonscientific subjective/biased/not-conclusive wording (especially in the introduction) that should be avoided. Some examples are "all areas" in line 27, "must be" in line 28, "absolutely crucial" in line 42, "existence of life itself" in line 43 and "ultimate frontier" in line 45.

This matter is very subjective (the other two reviewers did not raise similar concerns), but we changed the wording in the following way that (hopefully) will sound less provocative:

"all areas" in line 27: wording changed to "many areas" (currently Page 3, line 39).

"must be" in line 28: wording changed to "Digitalization of our environment requires an ever growing number of objects to be identified..." (currently Page 3, line 40).

"absolutely crucial" in line 42: word "absolutely" was deleted and the paragraph was slightly expanded and rearranged to better express what we wanted to convey to the reader (currently Page 4, lines 57-61).

"existence of life itself" in line 43: same as above, we expanded and rearranged this paragraph. However, in this case we decided to keep the phrase "existence of life itself", because we do not consider this as non-scientific or biased. Our intention was to stress the importance of information encoding at the molecular level in general, because it is not something that we encounter in our daily lives (though they depend on it). We continue with the next sentence to explain: "All known life forms depend on nucleic acids, which are in essence a digital medium." (Page 4, lines 57-59).

"ultimate frontier" in line 45: wording removed and sentence changed (currently Page 4, lines 67-69).

6. Lines 46-47 of the introduction describe DNA-based sensors for encoding digital data. Nevertheless, other examples based on different types of molecules should be given.

We have added to the first paragraph in introduction a sentence "Thus, alternative synthetic polymers have been proposed for information encoding" and we include relevant references (currently Page 4, lines 63-64).

We have previously provided references to some of such polymers in the introduction, in the first sentence of paragraph "*Writing* (encoding) information at the molecular level can be achieved with either (i) sequences of monomers concatenated within one molecular string" (currently Page 4, line 71).

7. Does the chirality of DO3A-Hyp affect the obtained results? Would you get different magnetic susceptibility tensors when using the R-Hyp isomer rather than a S-Hyp one? Would the chirality affect the PCS of a single lanthanide? An example would be very beneficial and a discussion of this issue is missing.

Yes, the PCS is very sensitive and different configuration on any of the three stereocenters of the Hyp moiety would result in very different PCS. To demonstrate this, we have synthesized a new analogue of the tripeptide molecule with one of the chelator building blocks being a different isomer (2*S*,3*R*,4*S* configuration, building block **L**²). This required significant amount of synthetic work and revision of synthetic schemes in the manuscript. The new isomer was also characterized by X-ray diffraction. Please see mainly the Fig. 6 for the results. The PCS of the new tripeptide **TP2** are very different from **TP1**. The results and discussion of this issue are distributed throughout the text:

(1) Pages 8-9, lines 145-159 ("Design of building blocks"). Here we discuss the different structures of **L**¹ and **L**².

(2) Page 10, lines 190-192 ("Encodable molecular framework") and Fig. 6. Here we discuss the different PCS effects provided by **TP1** and **TP2**.

(3) Page 12, lines 233-237 ("Rules of paramagnetic encoding"). Here we discuss that modification to the building block (difference between **L**¹ and **L**²) provides an opportunity to manipulate and optimize the resulting paramagnetic shifts.

8. The authors mention that one of the advantages of their approach (and maybe of other MRI-based approaches) over MS-based strategies is the ability to recycle the probe. Would it be possible to recycle it? Would one be able to separate between different probes located in a single mixture?

This is a misunderstanding. We did not suggest recycling of the probes, but claimed that the encoded information was repeatedly readable by magnetic resonance, because it is a non-destructive method (as opposed e.g. to mass spectrometry, DNA sequencing). Our practical experience in handling these molecules is that they would be nearly impossible to separate from mutual mixtures due to their chemical similarity. In fact, this is a distinctive and potentially advantageous feature of our molecular system in comparison with other approaches of encoding information. For example, chemically different molecules can be subjected to LC-MS analysis, where they would be separated from each other and distinguished by their molecular weight. The differences between our molecules encoded with metal ions are so small that it would be extremely difficult to separate them chromatographically. Moreover, positional isomers (e.g. HoDy-TP1 / DyHo-TP1) could not be distinguished by molecular weight. In combination, this makes our molecular system practically indecipherable by other techniques than magnetic resonance. We briefly discuss this in the newly added section "Digital information encoding and reading" (Page 16, lines 317-321)

9. In theory, there are many more combinations possible compared to the ones that can be practically used. The authors should discuss issues of spectral resolution and how this parameter affects the capability of the molecular system proposed.

A short discussion to this is provided:

(1) Page 12, lines 233-237 ("Rules of paramagnetic encoding"). Here we discuss how the individual paramagnetic contributions can be optimized via building block design to improve spectral resolution. The newly added **L**² building block and **TP2** framework provide good evidence that this is possible.

(2) Pages 14-15, lines 290-296 ("Multiplexing and combinatorial implications"). Here we talk about what is the current capability and what might be achieved with further optimization of the molecular system and method of reading.

10. Is an 8-bit code indeed applicable? For example, DyDy shows a chemical shift of -57.40 ppm, and TmTm of -57.61 (Table Supplementary Figure 9). Or, perhaps, this table is just not properly explained.

Yes, and more than that: **16-bit codes** are possible with the current system. The table in Supplementary Figure 9 (in this revision renumbered as Supplementary Fig. 18) is not the right place to look. Please see the tables in Supplementary Figures 8 and 10 (in this revision Supplementary Figs. 17 and 19) that highlight 4 different lanthanides, where all 16 permutations within the **TP1** molecule provide non-overlapping signals usable for multiplexing. These permutations allow 16-bit encoding. To show a practical example, we report an additional NMR experiment in this revision. Please see the new section "Digital information encoding and reading" (Pages 15-16) and the associated Fig. 8. Here we demonstrate encoding of a 10-letter password (in a binary code of ASCII characters) within a single NMR sample readable with a single z-resolved NMR measurement.

It is worth noting that some of the **M¹M²-TP1** molecules used in this experiment were synthesized only after this Reviewer's request to see a practical example. Nevertheless, they confirmed our original predictions of their NMR shifts. Please see the Supplementary Fig. 19, where we demonstrate successful decoding even of the new **M¹M²-TP1** molecules, while using the original predicted shifts (from the first submitted version of the manuscript).

Reviewer #2 (Remarks to the Author):

The article reports the design, synthesis, and characterization of a bimetallic system that can act as a molecular versatile platform for information encoding, readable by MRI. While the size of current information storage devices is reaching physical limitations, the concept of encoding and decoding information at the molecular level has never been more relevant. The presented manuscript proposes an elegant approach to this challenge, well supported by a sound and well-explained methodology.

11. Unfortunately, the principle, described here as unprecedented, is very similar to a very recent study published in *Nature Communication* (*Versatile non-luminescent color palette based on guest exchange dynamics in paramagnetic cavities*) by Amnon Bar-Shir and co-workers. doi: 10.1038/s41467-021-23179-9).

The recent work of Amnon Bar-Shir is indeed very impressive, and we have added a reference to it at several places in the manuscript. A fundamental difference between our work and previous works (including this one of Bar-Shir) is in the number of lanthanide ions present in each molecule. Previous works used a single lanthanide ion per molecule to generate the paramagnetic shifts. The limitations of this approach are the same as for codes based on luminescence or isotope mass, as we explain in the section "Multiplexing and combinatorial implications" and Fig. 7A. Our approach of combining two lanthanide ions in one molecule provides much higher number of combination and control over the resulting NMR shifts. We provide better explanation of this difference at these places:

- (1) Page 6, lines 111-123 (Introduction).
- (2) Page 11, lines 205-208 ("Rules of paramagnetic encoding").
- (3) Page 14, lines 274-279 ("Multiplexing and combinatorial implications"), starting with sentence "Notably, this applies also to multiplexing NMR signals..."
- (4) Page 16, lines 328-335 ("Parallel reading with MRI"), starting with "Spectrally resolved MRI..."

Following are my suggestions and comments on the presented manuscript:

12. - The introduction would benefit from a general scheme explaining the principles of the system and approach reported in this manuscript.

We thank the Reviewer for this suggestion. We have added a general scheme as Fig. 1, which is referenced from the Introduction (Page 6, line 123).

13. - The use of the term "monomer building block" is ambiguous as to what precise molecule it refers to. I initially understood the term for the Dy complex (monomer and dimers

usually used for the number of metallic centers in the molecules). If I am not mistaking, it refers to the ligand L^1 that is used as a synthon in the different synthetic approaches. The manuscript would benefit from further precision at the beginning of the paragraph.

We have removed the term "monomer" and only refer to our synthons as "building blocks". The building blocks are defined in Fig. 3 as differently protected variants of L^1 or L^2 . When necessary to stress that metal complexes were used as synthons, we refer to them as metal-preloaded (Page 10, line 182). We hope this clarifies the matter.

14. - Page 7, " ^{19}F NMR spectra showed four major singlets". Do you have a hypothesis on why there are more signals present? On the Supp. Fig. 3, are all integrations equal to 1 (especially for Nd/Tb)?

Here, the Reviewer refers to the NMR spectra of the statistical mixtures (previously Supplementary Fig. 3, in this revision Supplementary Fig. 9). It is important to note that these statistical mixtures were used without purification, and this fact was stated in the figure captions. Thus, lower purity can be reasonably expected. Skipping the purification step allowed us to screen many metal ion combinations much faster. It is also important to note that this approach worked well, because there was no problem in identifying the four main products, which then served to successfully determine the individual paramagnetic contributions of M^1 and M^2 in $M^1M^2\text{-TP1}$ system. We consider it quite important that these parameters could be "calibrated" using such a "quick and dirty" approach to save time and sample.

That said, we did not attempt to characterize the minor species observed in the spectra, because it would defeat the point to perform a quick screening. We expect that some of the peaks may correspond to byproducts arising from incomplete complexation of **TP1** with the metal ions. The likely reason is a pipetting error in preparation of the statistical mixtures, which were composed of 95 microliters of **TP1** stock solution and 2.5 microliters of stock solutions of each metal (Supplementary Information, Page 48, "Post-synthesis of $M^1M^2\text{-TP1}$ statistical mixtures"). Small errors thus may have caused in some cases molar deficiency of the metal ions, giving rise to impurities with one chelator unit empty.

However, some minor peaks were observed also in ^{19}F spectra of well-defined individually prepared compounds that were purified by preparative HPLC (e.g. Fig 6B). Attempts to further purify the compounds and remove the minor peaks were unsuccessful. Therefore, we assumed that these peaks probably represent some minor isomers or conformers in equilibrium with the main peak species (in slow exchange on NMR timescale). To test this hypothesis, we performed additional NMR measurements at elevated temperatures. Indeed, all but one of the minor peaks coalesced, confirming this hypothesis. We assume that also the last remaining minor peak is some other form of the main species, but with higher energy barrier for the exchange that prevents reaching coalescence at reasonable temperatures. To reflect on the occurrence of the minor peaks in the purified compounds, we included a short commentary in the "Encodable molecular framework" section (Page 10, lines 185-188) and added the NMR coalescence experiment as Supplementary Fig. 10.

Finally, regarding the question about integration. Yes, we checked the integration of the four major peaks in the Nd/Tb statistical mixture and all four give identical integrals (1.0 with decimal point precision). In some combinations, the peaks may look very different because of variable T_2 relaxation times. Short T_2 values caused by some of the paramagnetic ions make the peaks significantly broader and hence apparently smaller (in height) than others. But the integrals are comparable with a good precision. Nevertheless, this was not important for the present work and we decided not to include the integrals in Supplementary Fig. 9 to avoid overcrowding of the displayed information.

15. - Please amend the first sentence of the 'Rules of paramagnetic encoding' paragraph in consequence of the recently published paper mentioned above.

We changed the wording in this paragraph in line with this suggestion, and to better explain that the novelty of our molecular system lies in combination of paramagnetic effects within one molecule (Page 11, lines 205-208).

16. - 'Sequence decoding' paragraph: I found it hard to follow the reason why a bias-free set of parameters needed to be obtained. I would suggest to the authors the addition of a sentence or two before the introduction of this part on why this is necessary for future applications. Although stated at the end of the paragraph, it would help with the flow of the manuscript.

We have expanded the explanation approximately in the middle of this paragraph (Page 13, lines 248-253), but found it unsuitable for placement at the beginning, because there we start with a broader perspective covering other methods of decoding. We hope that the flow of text has improved.

Minor Details:

17. - Main text page 22, I am not convinced by the formulation of the complex as the asymmetric unit formula $[\text{Dy}(\text{L}^1)]_2 \cdot 7\text{H}_2\text{O}$ as it implies a dimerization, which is not the case here. I suggest describing either the repeating unit $[\text{Dy}(\text{L}^1)] \cdot 3.5\text{H}_2\text{O}$ or marking the "polymerization" as $[\text{Dy}(\text{L}^1)]_n \cdot 3.5\text{H}_2\text{O}$ (with a personal preference for the second notation). We changed the description according to this suggestions.

18. - ESI page 5, '*direct methods*' should be replaced by '*intrinsic phasing*', which is ShelXT (direct methods would be ShelXS).

Fixed. Now in Supplementary Information, page 6.

19. - ESI page 5 and Main text page 23: Add the unit Å to the Mo wavelength.

Fixed. Now in Supplementary Information, page 6.

20. - ESI page 7, '*D. Rearangement*' should read '*D. Rearrangement*'

Fixed. Now in Supplementary Information, page 14 (Supplementary Fig. 8).

X-Ray Diffraction study:

After careful examination of the crystal structure obtained and the chemicals involved in the synthesis, I do not have a better interpretation than the one proposed by the authors regarding the high residual density Q-peak located near the macrocycles. Therefore, I recommend the publication of the CIF file as presented.

Unfortunately, the newly added structure of $[\text{Dy}(\text{L}^2)] \cdot 3\text{H}_2\text{O}$ suffered from similar problems, and we provide similar explanation. This was despite of trying several different crystals.

21. However, the description of the rather complicated structure would benefit from a slight reorganization. I always find it clearer when the structure is described "increasing the scale" with corresponding figures associated with each step: I would suggest starting with the description of the mono-nuclear complex on its own, then the asymmetric unit where two complexes are present, followed by the helix 3D arrangement, then packing of the two helixes. Maybe the term "repeating unit" could be used from the polymer lexicon? Figures could be made clearer by using wireframe representation of the carbon atoms (except in the first description of the complex where ellipsoids are necessary).

We followed this advice and show the crystal structure at three levels of increasing scale, using ellipsoids for the lowest scale and the rest in wireframe. Please see Fig. 4, and Supplementary Figs. 1-3 for $[\text{Dy}(\text{L}^1)] \cdot 3.5\text{H}_2\text{O}$ and Supplementary Figs. 4-5 for the new structure of $[\text{Dy}(\text{L}^2)] \cdot 3\text{H}_2\text{O}$.

Reviewer #3 (Remarks to the Author):

This is a very interesting and well-executed work that presents a novel and exciting concept: The use of paramagnetic lanthanides to encode information at the molecular level. For this purpose the authors designed a ligand scaffold containing an amino acid moiety that can be used to join together two different paramagnetic lanthanide ions and a reporter unit containing a CF₃ group. The topic is introduced in a very clear way that non-specialists can follow. The manuscript contains a solid piece of experimental work, including the synthesis of the ligands, NMR spectroscopy and imaging and the characterization of the complexes using X-ray measurements.

For these reasons I support publication of this paper in Nat. Commun. after addressing a few minor issues:

22. -Page 4: "This tensor has radial and angular dependencies specific for each element". It is not actually only the nature of the element (lanthanide), but the identity of the system as a whole (the complex) that dictates the magnetic anisotropy, and thus the pseudocontact shift. Please rephrase.

Wording changed to: "This tensor has specific radial and angular dependencies dictated by the particular element and its coordination environment..." (currently Page 6, lines 106-109).

23. -Page 7: "The chemical shift induced by paramagnetic lanthanide ions cannot be predicted from theory with confidence". I agree with the authors that this is not a trivial task, but quite some progress was made over the last few years towards this direction. In particular wave function calculations were shown to predict reasonably well the magnetic anisotropies, and thus pseudocontact shifts, of lanthanide complexes (i. e. Yb(III) derivatives).

We changed the wording to "The chemical shifts induced by paramagnetic lanthanide ions are difficult to predict from theory with confidence, especially if the structure in solution is not precisely known." (currently Page 9, lines 172-173). This is a milder and more accurate expression in line with the Reviewer's comment. However, we did not want to elaborate on the theory too much, as it is not important for our work, which relies on experimentally determined shifts. Interested reader will be able to find relevant sources through the references.

24. - Rules of paramagnetic encoding: The analysis of the paramagnetic shifts presented in the paper is very interesting. The ¹⁹F nucleus is rather far away from the paramagnetic center in terms of number of bonds, and thus contact contributions are expected to be negligible. I miss here some comment on the origin of the paramagnetic shifts and their comparison with the predictions made by Bleaney's theory. This theory has been recently put into question by different authors, but still represents an useful guide to design lanthanide paramagnetic systems. I have plotted the paramagnetic contributions shown at the bottom of Supplementary Fig. 7 versus the Bleaney constants. One observes a rather good linear correlation, except for a couple of points that fall off the linear trend. The qualitative trend observed here is nevertheless well reproduced by Bleaney's theory. I understand that the aim of this paper is not to provide a detailed theoretical analysis, but still some comment along these lines is welcome, as it would provide some rational (and confidence) to the analysis.

We agree with the Reviewer's observations and comments. Indeed, we have tested the correlation between the determined paramagnetic shift contributions and Bleaney constants and originally considered including it in the manuscript. The correlation is reasonably good, but some data points are clearly outside of the expected trend. It was unclear to us whether showing this imperfect correlation with (imperfect) Bleaney's theory is going to be useful. However, after this Reviewer's comment, we decided to add a short discussion to "Rules of paramagnetic encoding" (Page 12, lines 230-232), and a Supplementary Fig. 16 showing the correlations with Bleaney constants.

25. - The authors should at least provide some comment on the origin of the minor signals present in the NMR spectra (Fig. 5). If avoiding different isomers in solution was a key objective, DO3A derivatives may not be the best choice. I admit however that the population in solution is large dominated by a major isomer, but this is something difficult to predict before synthesizing and characterizing the complexes.

Please see our detailed response to a similar question asked by Reviewer 2 above (point 14.). A comment was added to the main text (Page 10, lines 185-187) and an additional NMR coalescence experiment provided in Supplementary Fig. 10. We agree that isomerism is difficult to predict, but the rationale behind the molecular design of building blocks **L**¹ and **L**² was good and we were fortunate that it worked as intended.

Reviewers' Comments:

Reviewer #1:

Remarks to the Author:

The authors have thoroughly addressed the comments of all three reviewers and improved their manuscript to my satisfaction. Having demonstrated several specific examples for information encoding/decoding using both 2D-NMR and MRI setups, the authors clearly showed the performances of the proposed approach that were missing in their first submission. I congratulate the authors for their achievements.

Reviewer #3:

Remarks to the Author:

The authors have addressed all comments provided in my previous report in a satisfactory manner. Furthermore, I am very impressed by the revision that the authors completed in response to the concerns raised by other two reviewers. In particular, I found very impressive the experiment presented in Fig. 8, in which password decoding is performed by NMR spectroscopy, using a sample in which solutions of different lanthanide pairs are arranged in aqueous layers separated by CCl₄. The incorporation of a new figure (Fig. 1) describing the principle presented in the manuscript improves the readability of the manuscript. The novelty of the work is well justified, with the use of two lanthanide ions providing a clear advantage over previous mononuclear systems. Advantages over alternative methods such as MS or optical methods are also clear, and the shortcomings discussed (NMR is certainly an expensive method). I have also checked the stereochemistry of the X-ray structures, which has been correctly assigned. In summary, this is a very impressive piece of work presenting a huge amount of data and a very new and interesting concept. I therefore support the publication of this paper in its current form.

Point-by-point response to the REVIEWER COMMENTS

for

second revision of the manuscript "Paramagnetic encoding of molecules"

Reviewer #1 (Remarks to the Author):

The authors have thoroughly addressed the comments of all three reviewers and improved their manuscript to my satisfaction. Having demonstrated several specific examples for information encoding/decoding using both 2D-NMR and MRI setups, the authors clearly showed the performances of the proposed approach that were missing in their first submission. I congratulate the authors for their achievements.

Our response:

There is no issue to discuss. We thank the Reviewer for all the suggestions and comments, which helped to improve this manuscript significantly.

Reviewer #3 (Remarks to the Author):

The authors have addressed all comments provided in my previous report in a satisfactory manner. Furthermore, I am very impressed by the revision that the authors completed in response to the concerns raised by other two reviewers. In particular, I found very impressive the experiment presented in Fig. 8, in which password decoding is performed by NMR spectroscopy, using a sample in which solutions of different lanthanide pairs are arranged in aqueous layers separated by CCl₄. The incorporation of a new figure (Fig. 1) describing the principle presented in the manuscript improves the readability of the manuscript. The novelty of the work is well justified, with the use of two lanthanide ions providing a clear advantage over previous mononuclear systems. Advantages over alternative methods such as MS or optical methods are also clear, and the shortcomings discussed (NMR is certainly an expensive method). I have also checked the stereochemistry of the X-ray structures, which has been correctly assigned. In summary, this is a very impressive piece of work presenting a huge amount of data and a very new and interesting concept. I therefore support the publication of this paper in its current form.

Our response:

There is no issue to discuss. We thank the Reviewer for all the suggestions and comments, which helped to improve this manuscript significantly.